# Values Education in Outdoor Environmental Education Programs from the Perspective of Practitioners

**Jan Činčera [1,*], Bruce Johnson [2], Roman Kroufek [3] and Petra Šimonová [1]**

1   Department of Environmental Studies, Faculty of Social Studies, Masaryk University, 60200 Brno, Czech Republic; simonova.tereza@gmail.com
2   College of Education, University of Arizona, 1430 E. Second Street, Tucson, AZ 85721, USA; brucej@arizona.edu
3   Department of Preschool and Primary Education, Faculty of Education, Jan Evangelista Purkyně University in Ústí nad Labem, 40096 Ústí nad Labem, Czech Republic; kroufek@gmail.com
*   Correspondence: honzacincera@gmail.com

**Abstract:** Shaping environmental values is considered one of the goals of environmental education. At the same time, this creates questions about the line between indoctrination and education. While values education has been widely discussed from various theoretical perspectives, few studies have analyzed how it is being practiced. This article investigates five outdoor environmental education programs and identifies the values the programs promote as well as the means they use to communicate these values to students. Additionally, the article examines the perspectives of 17 program leaders and center directors regarding the ways in which values should be promoted in environmental education and the approaches they use in their practice. According to the findings, all the observed programs applied a normative, value-laden approach, communicating mainly the values of universalism. The most frequently observed strategy was the inculcation of desirable values by moralizing and modeling. Simultaneously, some of the leaders' beliefs, while highlighting value-free or pluralistic approaches, contradicted their rather normative practice. This article describes the theory–practice gap identified and discusses the implications of the prevailing use of the normative approach in outdoor environmental education for the field. It calls for opening an in-depth debate on what, why, and how values belong in outdoor environmental education practice.

**Keywords:** values; environmental education; outdoor programs; qualitative

## 1. Introduction

Should environmental educators promote particular values in their programs, or should they strive to be value-free? Alternatively, should these educators try to find a pluralistic approach, balancing various, even contradictory, values in their programs? This study investigates the challenging questions associated with reflecting values in the practice of outdoor environmental education. Particularly, we discuss three possible approaches and the related ways in which they are reflected and applied by program leaders: a value-free approach (no values should be promoted), a normative approach (particular values should be promoted), and a pluralistic approach (different values should be compared and critically reflected on).

Traditionally, shaping environmental values is considered one of the essential goals of environmental education. Environmental education is supposed to develop students' worldview, moral perspectives, and values in such a way that promotes students' motivation and willingness to participate in environmentally responsible actions on both individual and collective levels [1–3].

Additionally, these aims seem to be crucial in creating the necessary social motivation to cause a social shift toward sustainability and the acceptance of collateral transition constrains. Many environmental education programs aim to change students' environmental values, and the effectiveness of these programs has been demonstrated in numerous studies [4–8]. On the other hand, certain other studies have found little or no influence on values [9].

At the same time, shaping environmental attitudes and values is one of the most strongly contested areas in environmental and sustainability education. Since the first theoretical reflections in the field appeared, many authors have pointed out the risks associated with promoting particular sets of values in education. Hug [10] provided probably the earliest guidelines for clarification regarding this issue when he sharply differentiated between "education" and "advocacy". According to Hug, while both positions may be useful in society, they also fundamentally differ from each other and cannot be merged: environmental educators should not be mediators of environmental messages. Instead, they should help students develop the skills necessary for active involvement in environmental issues or help students to clarify their own values.

In spite of Hug's assertion, environmental values have remained a major part of environmental education programs. Decades of research have shown that affective variables such as environmental sensitivity, eco-centric values, and attitudes play an important role in people's willingness to engage in pro-environmental behavior [11,12]. In light of this, value-free environmental education would contradict its fundamental principles.

In addition, as many scholars argue, education is never value-free [13] and is always rooted in implicit or explicit values [14–17]. As Veugelers [16] suggests, teachers may declare their value neutrality, but it is never fully achieved because all teachers express particular values in their teaching. Besides, other values are communicated indirectly, through hidden curricula (for example, the level of students' freedom to participate in decision-making in their school) or prescribed curricula (for example, the values of democracy or growth that are often incorporated into national curricula documents).

While education may never be value-free, environmental education, being closely associated with a critical approach toward the practices of the dominant society, may support values that could be perceived as different from or at least not broadly supported within the dominant society. This leads to tensions reflected in both external criticism and internal debates within the field. In the United States, it was Sanera [18] who led such criticism of environmental education, arguing that it should focus strictly on facts and should not communicate particular attitudes. However, the necessity to find a balance between indoctrination (i.e., teaching students to uncritically accept particular environmental attitudes or values) and education has been discussed by many authors in the scientific community [19–23]. Other tensions have emerged from various ethical perspectives reflected by particular approaches existing in the field, particularly between the proponents of education for sustainable development and those who promote bio-centric approaches in environmental education [21,24].

In light of this, the question of how to deal with values in environmental education may be more important than why.

Caduto [25] identified eight distinctive strategies that can be applied in environmental values education:

- laissez-faire (not dealing with values at all),
- moral development (focusing on gradual age-appropriate methods for dealing with issues by discussing ethical dilemmas with students),
- inculcation (instilling particular values considered to be desirable through moralizing, modeling, or reinforcement),
- values analyses (based on logical thinking and analyzing values),
- value clarification (highlighting students' self-awareness and the identification of their values),
- service learning (based on learning values through action),
- behavior modification (assuming value change after behavior change), and

- confluent education (based on a holistic approach in schools).

Caduto also pointed out that environmental values education must take into account the age of the students. For young students, it is appropriate to inculcate some basic, positive social and environmental values. At the age of eleven to twelve, students start to be emotionally autonomous and educators should act more as guides that help students understand their own personal values as well as the values of others [25].

Other authors stress the importance of fairness in issue analyses, based on opportunities to assess various perspectives, consistency with democratic principles, and critical attention to the role of language in communicating environmental values [14,26]. A significant role in developing the environmental values of young students is played by outdoor experiences that provide them with opportunities to develop empathy towards the non-human world [27–29].

The place of outdoor education in the formal education curriculum of different countries varies greatly and is discussed quite often on a national level [30–33]. Although outdoor education is used with increasing frequency, it is still common for it be included with different levels of intensity at different stages of school attendance. While in kindergartens and primary schools, outdoor education is usually an integral part of education every day, in secondary schools it is employed relatively rarely.

However, the practice of environmental education programs often suffers from poor theoretical support when it comes to intervention strategies [34]. The existing research and evaluation studies usually focus on program outcomes, rarely analyzing particular intervention strategies and their meaning for students. As a result, while we know whether particular programs are effective or not, it is difficult to identify the instructional strategies that promote environmental values. To overcome the theory–practice gap, a network of seven organizations representing national networks that promote outdoor and environmental education was established in 2013. Besides the founding organizations from the United Kingdom, Germany, Italy, the Czech Republic, Hungary, and Slovenia, there were also more than thirty other partners, including centers and universities. The main output of the network was The Real World Learning (RWL) Model, which was published after more than two years of discussions among all the partners [35].

While the model does not provide a comprehensive definition of outdoor environmental education, it highlights the practice of outdoor education focused on sustainability. Traditionally, Priest [36] differentiated between two approaches to outdoor education: a) adventure education, which is focused on intra- and inter-personal relationships; and b) environmental education, which aims to take learning environmental concepts outdoors. To make it clear that the focus of this study is on the latter approach, we use the term "outdoor environmental education".

According to this model, outdoor environmental education programs (OEEPs) should promote self-transcendence values (i.e., the values of universalism and benevolence) as they are defined by Schwarz [37,38]. The model further assumes that these values should be promoted experientially:

> If we can develop those values that promote sustainable ways of thinking and being through our work, then we are on the right path. It is essential that we offer the learners a chance to live these values through the learning experience. It has been shown that values affect behavior, and also that repeated behaviors affect values. Teaching values does not change values; living values leads to values being changed and reinforced [39].

In particular, the model highlights the importance of three core values: respect for nature and caring about the state of our planet, equal opportunities for all people to shape their lives, and respect for future generations [39]. It is clear that the model supports the concept of normative rather than pluralistic or value-free outdoor environmental education. According to Schwarz [37,38], promoting particular values implies compromising other values. For example, by promoting the values of universalism (see Table 2), we compromise the categories of values connected with achievement and power (self-enhancement).

The question of applying a normative, pluralistic, or value-free approach has its consequences. While the normative approach can be considered a straightforward way to achieve the aims of

environmental education [1–3], it may also cause tensions between the programs and students who favor other values (for example, power, security, achievement). A similar tension may emerge between the programs and the students' families. While intergenerational transfer of environmental education effects is usually described as non-problematic or only slightly problematic [40,41], there is some evidence of such tension in several papers on the topic. Johnson and Cincera [8] reported a clash between students who wanted to change their practice after participating in an OEEP and their parents, who rejected such attempts as useless and foolish. A similar issue was described by Chineka and Yasukawa [42] in their study of the attempts of EcoTeam members to change their home agriculture practice in Zimbabwe.

Given the theory–practice gap [34], it is worth investigating the ways in which values are being developed in the practice of OEEPs. Particularly, we will focus on the following questions:

- What do program leaders think about the importance of values education in OEEPs? What are the theories of sound values education in OEEPs?
- How do program leaders promote values development in their OEEPs? What particular values do they communicate?

## 2. Materials and Methods

To answer the above questions, we combined two methods of obtaining data. First, we selected five OEEPs for elementary school students in the Czech Republic. A description of the programs can be found in Table 1 (the names of the programs have been anonymized).

**Table 1.** A description of the analyzed programs.

| Program Code | Program Description |
| --- | --- |
| Yellow | A 5-day residential program in a rural environment. Focuses on developing outdoor skills (orientation in nature, starting a fire, cooking at a campfire, etc.) and affinity toward nature. |
| Green | A 5-day residential program in a wetland area. Focuses on developing environmental attitudes and ecological understanding. |
| Orange | A 3-day residential program in a sandstone rock area, with follow-up activities. Focuses on developing environmental attitudes, ecological understanding, and behavior change. |
| Blue | A 3-day residential program in a mountainous area. Focuses on interpretation of the natural heritage of, and developing a relationship to, a protected locality in the mountainous area. |
| White | A 5-day residential program in a karst area. Focuses on developing outdoor skills and encourages spending time in nature. |

Two observers independently observed each of the programs. The observers noted down all the instances of the program leaders' communicating a value-laden message to the students. The data segments were further categorized into a scheme derived from Schwarz's [37,38] theory of universal values (see Table 2).

For example, when a program leader told the students a personal story about watching animals and commented on their beauty, the observer categorized this instance as "world of beauty" in the core category "universalism". Similarly, when a program leader encouraged the students to enjoy direct and stimulating contact with nature, the observer categorized this instance as "excitement in life" in the core category "stimulation".

Each observer analyzed his or her observations independently. After this, they compared their analyses to avoid possible misinterpretation and identify the prevailing features of the programs.

**Table 2.** Categorization of the observed values.

| Core Values | Specific Values |
| --- | --- |
| Universalism | Broadminded, equality, unity with the world, protecting the environment, a world of beauty, inner harmony, a world peace, social justice, wisdom. |
| Self-direction | Freedom, independent, curious, creativity, choosing own goals, privacy, self-respect. |
| Benevolence | Mature love, spiritual life, helpful, forgiving, true friendship, the meaning of life, honest, responsible, loyal. |
| Stimulation | Daring, variation in life, excitement in life. |
| Hedonism | Enjoying life, self-indulgent, pleasure. |
| Conformity | Self-discipline, politeness, honoring elders, obedient. |
| Tradition | Humble, detachment, respect for tradition, devout, moderate, accepting my portion in life. |
| Achievement | Intelligent, capable, successful, influential, ambitious. |
| Power | Social recognition, social power, wealth, authority, preserving my public image. |
| Security | Healthy, family security, social order, clean, sense of belonging, reciprocation of favors, national security. |

Further, the observers noted down all instances of the program leaders attributing an intrinsic (bio-centric) or instrumental (anthropocentric) value to nature. For the observations, we used Bogners' [43] definitions of environmental values, i.e., preservation of nature (a bio-centric perspective on the conservation and protection of the environment), utilization of nature (an anthropocentric perspective on the utilization of nature), or appreciation of nature (a disposition to perceive a positive experience of nature in nature settings).

Preservation of nature was coded in the instances when the program leaders communicated that nature should be protected because of its inner (for example, eco-systemic) qualities. Utilization of nature was coded when the program leaders communicated that nature is important for satisfying human material needs. Appreciation of nature was coded when the program leaders communicated that nature is a source of non-materialistic benefits (such as enjoyment).

Additionally, we interviewed the program leaders for each of the programs. Altogether, we got responses from fifteen program leaders and two directors of the program centers. Of the seventeen respondents, nine were female and eight were male. We tried to investigate all the leaders who were actively engaged in the analyzed programs. Due to the different size of each center, the number of respondents for each program varied between two (Blue program) to four (Green, Yellow, and Orange programs). There were also differences among the respondents in terms of their practice, ranging from "junior" leaders (less than 5 years of practice) to highly experienced "senior" leaders (more than 15 years of practice). Moreover, the respondents differed in their "life-path" to environmental education. Some of them were attracted to it during their university pre-service teacher training and started their career at the center. Others had no formal teacher training and graduated from other fields relevant to environmental education. None of the respondents were staying at the center; they all commuted from nearby towns. These differences should be considered a limitation of the analyses, but they offer a helpful illustration of the reality of the educational settings at the centers.

The interviews were part of a larger project in which the program leaders of the selected OEEPs were asked for their opinions regarding different instructional strategies applied in OEEPs. To examine the communicating of values, we asked the following questions:

According to some researchers, program leaders may influence the ways in which the program impacts students through the values they communicate. For example, if we say that nature should be well managed, we highlight values connected with power and achievement, which may decrease the chance of promoting pro-environmental behavior. Other people believe

that this is unimportant or even reasonable—nature should be managed by humankind. What do you think about this?

How is it in your program? Are there any particular values you would like to communicate?

We used semi-structured interviews to allow the interviewers to be flexible and to be able to follow up on ideas shared by the respondents. For example, when a respondent mentioned that the program should communicate particular values but not force them on the students, the interviewer asked the respondent what they think about how their program deals with this. When a respondent mentioned that the program should communicate "good" values, the interviewer asked for clarification on what these "good" values are that should be communicated.

Generally, the interviewers tried to encourage the respondents to express any opinion on this topic, without indicating that some responses are more desirable than others. All of the responses were audio-recorded and transcribed. The respondents' ideas were further categorized into three main categories: a) the value-free approach (the program should not promote any particular values), b) the pluralistic approach (the program should encourage students to analyze and compare different values), and c) the normative approach (the program may support a particular set of values), with more subcategories in the last group (for example, naturally from experience, education is manipulation, etc.).

This methodology has its limitations. As we only observed five different programs, the findings cannot be generalized. Despite our effort to ensure the validity of the observations, the programs could have been affected by the unusual conditions of being observed, which may have altered the program leaders' work. In addition, some of the programs tended to be "fluid", changing according to the particular leader who was involved at the time. As a result, the findings cannot be interpreted primarily as information about the observed programs but rather as an analysis of the ways program leaders deal with values education in OEEPs.

## 3. Results

In our sample, most of the respondents believed that the question of values education in OEEPs is highly important. Altogether, we were able to identify three distinct approaches, ranging from value-free to pluralistic and value-laden. In the following text, we describe each approach from the perspective of the leaders' beliefs as described in the interviews and then from the perspective of the leaders' practice as we observed it.

### 3.1. The Value-Free Approach

#### 3.1.1. Leaders' Beliefs

The belief that OEEPs should be value-free almost did not appear in our sample. Only two of the respondents said they held this belief. As they saw it, program leaders should avoid any intentional values education in their programs and instead allow the students to shape their values on their own, based on their interpretation of what they experience during the program:

When I communicate with the children, I try not to transmit the values ( . . . ) rather, I try to describe what a thing looks like or its factual aspects. (K, leader, Green program)

I do not think we should push any values into children's heads ( . . . ) I like when they, after what they have seen and done, make their own values. (G, leader, Blue program)

#### 3.1.2. Leaders' Practice

Based on our observations, none of the programs was value-free. In all of them, the observers identified particular values that were communicated by the main theme (frame) of the program or by the program leaders. Both of the respondents who said they held a value-free belief repeatedly expressed particular values when presenting environmental concepts to their students.

For example, leader G (Blue program) taught students that "animals also have the right to have breakfast, like we do" (protection of nature, unity with the world) or that "peat bogs are important to us because they provide us with an enjoyable experience" (appreciation of nature, excitement in life). In these outdoor lessons, G followed the guidelines for the Blue program, communicating various perspectives on valuing the protected area of peat bogs.

However, G also assumed that the Blue program corresponded with his belief in a value-free approach and was unaware of the existing contradiction. Similarly, while leader K (Green program) reported that he tried not to transmit particular values to his students, the program that he led was, as we illustrate below, clearly value-based.

In light of this, it can be argued that while the practice of both of these respondents contradicted their beliefs, they did not reflect on this contradiction and did not seem conscious of it.

*3.2. The Pluralistic Approach*

3.2.1. Leaders' Beliefs

Three other respondents believed that program leaders should introduce students to various values or value-based interpretations. For example, U (leader, Orange program) believes that when teaching about a forest, program leaders should explain to the students that it can be interpreted as a source of both instrumental and intrinsic value:

> ( … ) to keep the balance between both poles ( … ) If we teach in the program just the first pole, or just the other, then it may happen that the children, when a little older, will hit the wall of this "truth" ( … ) and then they will say, "they were lying to us".

Another respondent, L (director, Green program) argued for involving the students in a story presenting different value positions and helping the students to analyze the weak and strong points of each position. Moreover, J (leader, Yellow program) believed that a group of leaders in the same program might not even share the same values. Therefore, the leaders should have the freedom to communicate the values important to them, even if it means the program will communicate different value perspectives. Similarly, O (leader, White program) believed that the process of value communication is unintentional, "it flows naturally from people" and "remains on the human rather than the leaders' level".

3.2.2. Leaders' Practice

Based on our observation, all of the observed programs communicated a variety of values. For example, the Yellow program communicated the values of universalism (protection of nature, unity with nature, wisdom), stimulation (daring), and achievement (success). Specifically, the leaders encouraged students to act carefully in nature ("Native Americans had the wisdom to live in harmony with nature") and to cooperate ("this is not a competition"). However, they also encouraged the values of achievement and success by counting the students' scores in selected activities and stimulating the students to achieve more ("you have got a high score … you can do this to get triple more").

In most of the programs, nature was associated with both instrumental values and intrinsic values. For example, in the Green program, the program leaders communicated the need for nature protection from an eco-centric position ("earthworms are very important to nature") as well as an anthropocentric position ("what seeds mean to us, how we can use them").

However, the opportunity for value analyses was not used because none of the leaders in the observed programs encouraged the students to reflect on the communicated values and analyze them from their own perspective.

As a result, while the students were exposed to different value perspectives, they did not have a real opportunity to grasp their differences and compare them with their own values. The pluralistic

approach, while accented by some of the respondents, remained limited to plurality in transmitted values and did not achieve a dialogical form.

## 3.3. The Normative Approach

### 3.3.1. Leaders' Beliefs

As most of the respondents reported, OEEPs shape or should shape students' values in accordance with the OEEPs' mission. However, the respondents differed in their opinions on how this should be done. The first group of respondents believed that while OEEPs influence students' values, this should be done naturally, based on the students' emerging experience, without planning. Nevertheless, these respondents also admitted that their own OEEPs were not value-free because they communicated particular values by their design, priorities, or goals. The respondents realized that some basic values, such as respect toward nature, animals, and nature protection, were so strongly incorporated into their OEEPs that the programs could not be value-free. For example, I (leader, Green program) comments:

> I do not see our program as value-free. It is about what it is made of, respect toward life ( … ) and toward what we investigate ( … ). And I do not think that the values are connected with nature only. It is also about how we speak about cooperation ( … ), that we do not use competition in this program, this is also value-laden to me. And also, how I speak about what I appreciate affects how the children perceive it.

Furthermore, the leaders indirectly promote particular values by modeling their interest in nature or accepting environmentally responsible behavior.

Similarly, another group of respondents accepted that they manipulate students, in subtle and socially desirable ways, toward particular values. According to them, education is always a kind of manipulation, and environmental education is no different. The key is to respect the participants' freedom, to communicate particular values but not to force students to accept them. According to Z (leader, Orange program),

> I do not think teachers should be value-free. On the contrary, teachers must be clear and readable in their values, but it is up to the children whether they accept these values or just some of them.

### 3.3.2. Leaders' Practice

Given that the observed programs did not provide an opportunity for students to analyze the communicated values, we can assume that all of them applied the normative approach. The most frequently communicated values were from the categories of universalism (protecting the environment, the world of beauty, unity with the world, inner harmony, wisdom), benevolence (cooperation), self-direction (independence, curiosity, choosing our goals), stimulation (daring), and hedonism (pleasure). Marginally, other values also emerged, such as achievement (success), power (authority), conformity (self-discipline), or security (sense of belonging, social order).

These values were inculcated through modeling, moralizing, and value-based language used by the program leaders. For example, L (leader, White program) applied a moralistic approach when an opportunity naturally emerged during a field trip:

> **L (leader, White program):** Shares a story about the environmental consequences of hunting deer in the area.

> **Boy:** And why should the deer be here, then?

> **L (leader):** And why are you here? They simply live here on their own.

In another situation, the same leader corrected the value-based statement of a student who commented on some copulating bugs she had noticed. His reaction directly motivated another student to express her view corresponding with the leaders' perspective:

**Girl 1:** Oh, these ugly bugs!

**L (leader):** They are useful. It would not work without them ...

**Girl 2:** They are cute!

In other cases, moralizing was used intentionally as part of program activities. For example, in the Orange program, students are taught to accept the values of protecting the environment and respecting nature. During an activity when the students were learning about the food chain, the program leader expressed the importance of these values:

**Leader:** We have to protect nature, so be careful and do not harm animals.

**Leader:** Think better about your decisions, it does not take much to disrupt food chains.

Modeling was another frequent strategy in the observed programs. For example, the program leaders in the White and Blue programs used the opportunity to pick up litter in the protected areas and encouraged the students to help them with it. In addition, the leaders in the White and Green programs modeled an authentic interest in nature observation, which was partly mimicked by the students:

**Leader (after the students found a big bug):** Oh, you beauty! Oh, it is so magnificent!

In other cases, the same leader used expressions such as "an amazing season" (world of beauty) or highlighted the importance of careful behavior toward animals ("the spider is also afraid, like us").

Further, values associated with protecting nature were indirectly communicated by the facilities of three of the programs, which were designed according to the principles of energy and water efficiency and which provided local, healthy, and mostly organic food. However, some of the students who found it unusual and not tasty rejected this sustainable food.

The last method of inculcation was the communication of particular values through the overall framing of the program. For example, the Orange program was introduced as a training center for students who want to become earth keepers. As a result, protection of the environment was the underlying value communicated throughout the program. Similarly, the Yellow program was framed with a story of Native American woodcraft. As a result, the value of Native American wisdom was repeatedly highlighted in the program's activities.

## 4. Discussion

Our findings have several important implications. First, while the discussion of shaping environmental values is ongoing in the field of environmental education, the practice of outdoor environmental education is based on inculcating values. None of the observed programs was value-free, and each of them influenced the students' value orientation. Such findings contrasted with the opinion of some of the interviewed program leaders who did not agree with this approach.

While we observed only five OEEPs, it seems likely that most OEEPs are value-laden, driven by the overall mission of environmental education. It is difficult to imagine an OEEP that would not communicate the values of nature protection, natural beauty, or other similar values. It is reasonable to say that a value-free OEEP would be a rare exception rather than a rule.

Concerning the relationship between education and indoctrination discussed in existing scholarly sources [19–23], we may perceive practice in OEEPs from both perspectives. Strictly put, OEEPs inculcate students with a particular set of values, mainly those from the category of universalism [37,38]. This is in agreement with the recommendation of the Real World Learning Model [35]. It may be

reasonable to argue that these values have been broadly accepted by society and that they form the core of environmental education [1–3]. Moreover, we did not observe any incidents of forcing particular values on the students. Instead, the students had a choice to accept the communicated values or not.

However, it would be hypocritical to ignore the fact that, when faced with particular values communicated through repeated moralizing and modeling by the program leaders, the students had no fair opportunity to defend any potentially contesting values. No matter how respectful and tolerant the observed program leaders were, they did not provide the students with the opportunity to challenge, analyze, or discuss the values presented by the program. Encouraging students to examine and discuss the values which underpin the relationships between humans and the environment should be one of the tasks of environmental education, too [44].

Moreover, based on Schwarz [37,38], by supporting some values, we compromise the other categories of values. This may create some tension in a society where self-enhancement values are accepted and, to a certain degree, promoted. In light of this, waves of criticism of environmental education, like those started by Sanera [18], seem inevitable. At the same time, outdoor education leaders should consider the implications for emerging tensions between the program and some of the participants or their families. These tensions may have further practical impact. For example, in the Czech Republic, it is up to the parents to decide whether their child will participate in an OEEP or not. When the parents suspect a possible clash between the values promoted by the OEEP and the values promoted by the family, they may not allow their child's participation, or they may compromise the program's impact when the child comes back. When the students perceive that the OEEP contradicts their values, they may refuse it, dislike it, or find an alternative agenda in the program (like socialization with their peers). As a result, values education in OEEPs may be accepted mostly by students sharing the same values, with a limited opportunity for their further promotion. This implication should be considered carefully. The positive effect of OEEPs on students' environmental values and attitudes has been reported by many studies [5–8,43]. However, it could be argued that the investigated effects might have been even stronger if a different strategy for dealing with values had been applied in the OEEPs.

Another interesting aspect of our findings is the gap between some of the respondents' articulated theoretical position and their leadership practice. This finding documents the persisting gap between theory and practice reported by Hungerford [34]. This situation reflects the final source of tensions associated with the issue of values. Some of the program leaders likely interpret their value-laden practice as value-free to avoid emerging cognitive dissonance. This opens the question of how to help program leaders grasp the theoretical background of their practice, reflect on it, and find a balance between what they think and what they do.

In light of this, it may be worth re-opening other questions regarding the what, why, and how of OEEPs. For example, what role should OEEPs play in achieving the socially accepted goals of environmental education? How should program leaders reflect on the values communicated in these programs with their students? Are inculcation strategies for shaping students' values legitimized by the social desirability of the promoted values? These questions are still to be answered.

## 5. Conclusions

Considering the methodological limitations of this study, we must interpret our findings cautiously. Nevertheless, the findings do suggest the existence of a theory–practice gap when it comes to dealing with environmental values in OEEPs. It is reasonable to suppose that the approach that prevails in current practice is based on the inculcation of a particular set of values that are not questioned or critically analyzed. Moralizing and modelling appear to be the most widespread strategies applied by the program leaders. In addition, it seems that the leaders are not always aware that their practice contradicts their teaching beliefs.

Moreover, these findings may be interpreted in light of the debate about the tension between education and indoctrination, a debate which is ongoing in the field of environmental and sustainability

education. From this perspective, the findings provide support for understanding outdoor environmental education as always value-laden, rooted in particular values and supported by their practice.

In light of this, we suggest accepting the value-laden nature of outdoor environmental education programs. At the same time, in order to build a bridge between theory and practice, we must open a discussion about the ways in which OEEPs deal with values.

To promote this discussion, further investigation of this topic is needed. In this study, we applied a relatively new and uncontested approach to analyzing the implementation strategies applied in OEEPs. However, it would be worth modifying this approach and obtain further, more in-depth analyses. Moreover, other relevant research studies may extend the investigation of dealing with values to more programs, particularly in countries with various socio-cultural environments.

**Author Contributions:** Conceptualization, J.C. and B.J.; methodology, J.C., B.J., and P.S.; formal analysis, J.C.; investigation, J.C. and P.S.; writing—original draft preparation, J.C. and R.K.; writing—review and editing, B.J. and R.K. All authors have read and agreed to the published version of the manuscript.

**Funding:** This is one of the outputs of the project Promoting Behavioral and Value Change through Outdoor Environmental Education, which is supported by grant no. GA18-15374S provided by the Czech Science Foundation.

**Acknowledgments:** This study is one of the set of studies analyzing different aspects of intervention strategies applied in five outdoor environmental education programs focusing on shaping the environmental values and behavior of young students in the Czech Republic. Because of this, all the studies share the same description of the green, orange, blue, white, and yellow programs. We are thankful to our partners for their willingness to cooperate and open their programs to in-depth analyses.

**Conflicts of Interest:** The authors declare no conflict of interest. The funders had no role in the design of the study; in the collection, analyses, or interpretation of data; in the writing of the manuscript, or in the decision to publish the results.

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
