# Peer review of "Values Education in Outdoor Environmental Education Programs from the Perspective of Practitioners"

_sustainability, doi:10.3390/su12114700_

Round 1

Reviewer 1 Report

The manuscript entitled “Values Education in Outdoor Environmental  Education Programs from the Perspective of the  Practitioners” could be of high interest for the readers of Sustainability journal. The role of OEEP in developing citizens who are environmentally oriented and committed to  environmental protection is not questionable. The manuscript is nicely written, and the study is clearly presented.

I have several comments and suggestions before considering this manuscript for publication:

Since the topic of the article focuses on outdoor environmental education (OEE), I recommend to the authors to provide some more information about the characteristics of this approach or at least the working definitions of OEE they used in this study. Does the concept ‘outdoor environmental education’ different from the traditional concept of ‘outdoor education’? If yes – what are the differences? If not – indicate that and explain your decision to use the term OEE.

The description of the methodology is clear and nicely detailed. Please define the interview type that were used in the study (unstructured ? semi=structured?) and why you decided to use this type?

You mentioned the need to distinguish between education and indoctrination twice : first in the introduction and second in the discussion. I did not find explicit findings that reflect or address this idea or necessity. Please clarify how the difference between education and indoctrination is connected to your specific results and/or rationale.

Author Response

Dear Reviewer,

Thank you for your positive feedback and useful comments on our manuscript. Based on your recommendation, we have done the following modifications:

  • we have discussed the differences between "outdoor education" and "outdoor environmental education", and provided a definition of "outdoor environmental education" (l. 123-128);
  • we have specified the type of the interviews;
  • we have briefly clarified the difference between "education" and "indoctrination" (l. 76-77).

Once again, thank you for your help.

Best regards,

Authors

Reviewer 2 Report

The manuscript fits well the scopus of the journal Sustainability. The topic of the manuscript is novelty and methodological approach of Authors is very interesting. I have only some minor recommendation for Authors:

Authors should emphasize the relevance of the topic and main findings of the study for concept of sustainability (in accordance to the focus of the journal). It should be added to the sections Introduction and Conclusion.

Authors should use brackets [] only for references. Another way of using brackets in the body of manuscript is very confusing.

Authors should consider the order of references cited in the body of manuscript. In its current form, the order is a bit confusing. Also, Authors should use only one relevant citation in order to support one idea. Using more than one citation for one idea seems not be professional (e.g. in lines 35, 37 etc.). Some citations are repeatedly used, it seems not OK. I suggest extending of the list of references in order to avoid using the same references for different ideas.  

Section Conclusion can be improved: Authors should more precisely present main scientific (original) findings of this study, which are novelty in the topic and which can be interesting for international readers of the journal.

Line 16: I miss here any short statement related to current knowledge gaps in the topic.

Line 21: I miss here a clear and brief presentation of main scientific findings (max two or three sentences).

Lines 22-24: Each of scientific paper discuss the topic. But, Authors should say what comes from the discussion of results.

Lines 28-30: Using this statement in the scientific paper is very unusual, I recommend to remove it.

Line 326 – Wrong citation (not following rules of the journal).

Author Response

Dear Reviewer,

Thank you for your positive feedback on our manuscript. Based on this, we have made the following improvements:

  • we have linked the topic of the manuscript more explicitly with the concept of sustainability in both Introduction and Discussion;
  • we have avoided using [] for other, then reference purposes;
  • we have extensively re-written the Conclusion (see l. 434-454);
  • we have extensively improved the Abstract to meet suggested changes;
  • we have corrected the wrong reference on l. 326.

We have discussed your suggested change for references in the text. However, we believe that the current form corresponds with the  "Instructions for Authors" for this journal (https://www.mdpi.com/journal/sustainability/instructions ). In light of this, we decided to keep it. I hope this explanation is acceptable to you. Thank you for your understanding.

Once again, thank you for your useful comments.

Best regards,

Authors

Reviewer 3 Report

The authors of this paper make several interesting points but it could do with some revision of the text to make a clearer argument abiut values and their place in environmental education.

My rules for an abstract are that it must cover the following points but not necessarily in this exact order.

  1. context and background explained,
  2. purpose/goal/aim/objective of the study,
  3. what was done and why,
  4. how it was done, 
  5. what were the main findings, and
  6. what are the theoretical/practical implementation of the findings

 The final written product should show several links across these points in the abstract.

In this paper the context is first explained around line 100 and the first few lines to this point are a bit disjointed. Avoiding indoctrination is important and is mentioned several times in a set of remarks about values and environmental education (EE). Hug is quoted extensively, then it is discussed whether EE is value free or not, and noted that education is not. Sanera is mentioned who has suggested that EE should always be taught as value free) and this part of the introduction finishes by quoting Casuto´s suggestion that eight different strategies can be found in EE.

at the moment there is not a good match in the points made in the introduction and then in the materials and methods, which actually covers some of the data as well. The analysis of the interviews seems cursory at best and the final discussion builds on EE using three main strategies - value free, pluralistic and the normative approach of which the second and third have barely been mentioned. I suggest that the introduction be rewritten so that the three strategies are evident from the outset. Extensive use is made of interview quotes in the results section and they are already used to support the strategies. To end with there are several points made in the discussion. Some of these could be construed as weaknesses in the study and included in the methods and the others (from line 315 onwards) return to the subject of values as practised in schools. Some of these points could belong to the introduction rather than the discussion or left out all together. This is also true of the concept universalism which pops up now and then without any conclusive points being made.

Several key ideas appear in the paper in all  sections, there are two questions in 115 to 130, and then there is a discussion which goes off in several new directions. 

in summary  - try to answer the question "What is being discussed iin this paper? Why?"

Author Response

Dear Reviewer,

Thank you for your inspiring comments. We hope we were able to meet most of them. There is a list of the modifications we have done based on your recommendations:

  • we have extensively modified the abstract;
  • we have prepared a new, introductory paragraph to make the focus of the manuscript and all the three analyzed strategies clear from the beginning (l.  32-39);
  • we have removed some of the parts from Discussion to Introduction (l. 139-154 and 405-410);
  • we have removed examples of data from Methodology (l. 174-178).

We hope that these changes helped to make the manuscript and its focus more clear. Once again, thank you for your useful ideas.

Best regards,

Authors 

Reviewer 4 Report

First of all, the article presents a related and interesting topic according to the interests of the magazine.

Having said this, I consider the introductory part to be well founded, although since we are talking about open-air education programmes that promote the development of environmental values, it could be interesting to analyse what the state of the question is regarding environmental education at present, and how it is approached from the different educational stages (its existence in the educational curriculum).

On the other hand, I consider it fundamental to know some more aspects about the leaders of each program, who make up the research sample. What is the total number of interviewees? It would be useful to know more about their socio-demographic characteristics.

As for the description of the results, I think it could have been more detailed and better organized. Very superficial information is reported.

Finally, I believe that the statements made in the section on discussions and conclusions are not very scientifically rigorous given that the study is backed up by a few interviews conducted. I believe that further analysis is needed to be able to infer these ideas. 

Author Response

Dear Reviewer,

Thank you for your inspiring comments. We have improved the manuscript to meet the suggested changes. There is a list of the modifications we made to meet your suggestions:

  • we have included information about the state of the question, regarding outdoor education (l. 106-111);
  • we have specified the number of respondents and provided their demographic characteristics  (l. 193-204);
  • we have re-organized the Findings by including new sub-headers and providing additional text and data (l. 253-267, l. 282-284, l. 302-305, 348-355, l. 368-369);
  • we have made changes in Discussion and re-written the Conclusion (l. 435-454). 

Once again, thank you for your effort and useful ideas. 

Best regards,

Authors

Round 2

Reviewer 1 Report

I am satisfied with the corrections made by the authors in the current version of the manuscript. I suggest to accept this current version for publication

Reviewer 4 Report

The authors superficially solved the shortcomings of the manuscript. I consider that the article presents very superficial results and a qualitative methodology with 17 interviews does not seem to me rigorous for publication in a journal indexed in journal citation reports.